# FMGTranDD: A Deception Detection Method Based on Spatiotemporal Facial Abnormal Emotional Changes

## Abstract

While multimodal deception detection methods improve detection efficiency, they inevitably introduce higher data collection and processing costs. Deceptive behavior is often accompanied by emotional fluctuations such as tension, anxiety, and guilt, which can lead to contradictory, inconsistent, or suppressed emotional expressions in individuals' facial expressions.This paper regards deceptive behavior detection as an abnormal signal recognition problem, aiming to capture abnormal features from regular behavior patterns. First, faces in videos are converted into a set of learnable facial emotion embedding sequences. Subsequently, a Time-LSTM-GCN module is proposed to model the spatiotemporal relationships between these facial emotion embedding sequences. The combined adversarial loss optimizes the decision boundary for deceptive behaviors. This loss function consists of two main components: first, semi-supervised learning of dominant facial emotions enhances the representational power of the embedding sequence; second, by comparing the similarity between embedding nodes with the same emotion (positive samples) and embedding nodes with different emotions (negative samples), the model is encouraged to capture both local structure within the sequence and global differences between sequences. Experimental results show that our new baseline model outperforms existing deception detection methods based on multimodal or multi-type features. Code is provided in the supplementary material.

## 1 Introduction

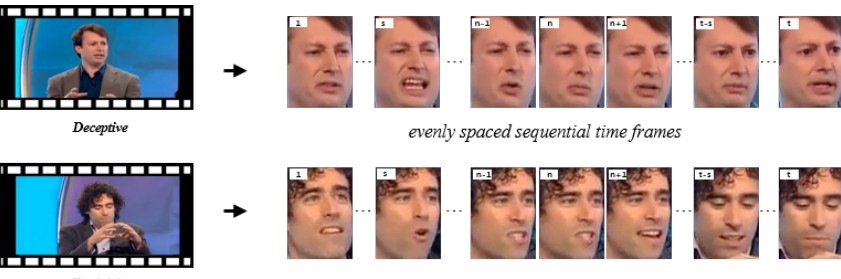

Figure 1: The process of facial expression changes between deceptive and non-deceptive behaviors.

With the rapid development and widespread application of digital media, ethical issues and security risks caused by human-centered deceptive behaviors are becoming increasingly prominent. Humans have a limited ability to detect deception, and without the aid of tools, their accuracy is only slightly above chance. Some behavioral cues may enhance its accuracy, but they are difficult for untrained people to detect (Wu et al., 2018). Automatic deception detection is a human-centric video analysis technique that has found applications in many real-world scenarios, including airport security checks, court trials, job interviews, and personal credit risk assessment (Ding et al., 2019).

Recent studies have explored multimodal approaches for integrating diverse deception cues (Wu et al., 2018; Krishnamurthy et al., 2023; Karimi et al., 2018; Karnati et al., 2022). However, collecting complete cross-modal data in real-world scenarios remains challenging, and inconsistent modality configurations across datasets hinder effective validation. Moreover, multimodal methods often incur high data costs. This motivates a deeper investigation into single-modality cues. Prior work (Ding et al., 2019; Zhu et al., 2025; Zhang et al., 2022; Khan et al., 2021; Yang et al., 2020) has shown that eye movements, lip dynamics, gestures, and body motions in the video modality provide rich contextual information, highlighting the potential of video-based deception detection.

Deception is influenced by subjective emotions and intentions, making it difficult to extract universal features across scenarios. Within the same context, deceptive and truthful behaviors often exhibit contrasting emotional patterns (see fig.1). Psychological studies show that when people conceal genuine emotions, involuntary micro-expressions may occur, such as stiff mouth corners, shifting gaze, or frequent blinking. Thus, deception is a dynamic process formed by subtle facial cues, which requires analyzing temporal evolution across frames rather than relying on a single static image.

This paper proposes a low-cost end-to-end automatic deception detection framework FMGTranDD. First, an automatic face acquisition strategy is adopted to systematically extract the sequential facial frames of the subjects in the video and generate the corresponding main emotion labels. Secondly, the Transformer facial representation encoder is used to extract basic facial features, map them into seven categories of emotion embeddings and construct time series. To distinguish between genuine and deceptive behaviors, we design a Time-LSTM-GCN module to model the spatiotemporal correlation features of sentiment embedding, and finally optimize the decision boundary of deceptive behaviors by combining adversarial losses. The main contributions of this work are:

- We designed a low-cost, end-to-end automatic deception detection framework, FMG-TranDD, based on LSTM-GCN. This framework automatically collects facial information and extracts facial representations, mapping them into facial emotion embedding sequences, thereby constructing a novel deception cue representation method.

- A Time-LSTM-GCN module is proposed to construct the spatiotemporal relationship of facial emotion sequences, and combined with a combined adversarial loss function to optimize the decision boundary of deceptive behavior.

- The experimental results show that the proposed new baseline model outperforms most of the existing methods using multi-modal or multi-type feature fusion in terms of performance indicators.

## 2 RELATED WORK

**Automatic Deception Detection.** In recent years, machine learning methods have significantly advanced the field of automatic deception detection. Early work, such as Krishnamurthy et al. (Krishnamurthy et al., 2023), proposed the first neural network-based model that leveraged multimodal features for deception detection. Wu et al. (Wu et al., 2018) systematically analyzed the importance of visual, audio, and textual modalities, while Karimi et al. (Karimi et al., 2018) introduced an end-to-end framework (DEV) to avoid the need for complex feature engineering. Subsequently, some studies shifted toward single-modality optimization. For example, Ding et al. (Ding et al., 2019) developed FFCSN focusing on facial and body features, Avola et al. (Avola et al., 2020) explored gesture features, and Yang et al. (Yang et al., 2020) proposed emotion transition features (ETF) for deception detection under limited data conditions. These methods highlight the contrast between truthful and deceptive behaviors but have yet to fully address the issue of cross-domain transferability. In terms of multimodal extensions, Karnati et al. (Karnati et al., 2022) incorporated EEG signals, Zhang et al. (Zhang et al., 2022) decomposed deceptive behaviors into question–answer units and proposed a graph-based cross-modal fusion model (GCFM), and Zhu et al. (Zhu et al., 2025) exploited inconsistencies among body parts to design a dynamic learning framework (DLF-BRAM). However, these approaches often rely on labor-intensive feature engineering. On the other hand, some studies focus on identifying key deception cues. For instance, Rill-García et al. (Rill-García et al., 2019) reported that gaze direction, eye landmarks, and acoustic features serve as stable indicators across cultures, while Khan et al. (Khan et al., 2021) emphasized the importance of eye and facial micro-movements as salient features for automatic deception detection.

Although existing research generally emphasizes the importance of multimodal fusion, it faces many limitations in cross-scenario applications. Current research on automatic deception detection often relies on multimodal features. However, in different scenarios (e.g., personal testimony(Gupta et al., 2019; Guo et al., 2023), interrogations(Xu et al., 2025), court trials(Pérez-Rosas et al., 2015)), the available modalities are not consistent, making it difficult for multimodal methods to generalize effectively. Moreover, multimodal approaches lack broad interpretability: although combining multiple modalities may appear to provide richer cues, deception signals are inherently hard to define, and errors in one modality can interfere with others. Therefore, this study focuses solely on the video modality from public datasets, analyzing emotional dynamics presented by subjects and assuming that truthful and deceptive behaviors exhibit contrasting emotional expressions.

**Abnormal Signal Detection.** In the field of anomaly detection, Long Short-Term Memory networks (LSTMs) and Graph Convolutional Networks (GCNs) have been widely adopted due to their strengths in temporal modeling and spatial relation learning. LSTMs are effective in capturing long-term dependencies in sequential data, and prior studies have applied them with OC-SVM or SVDD for unsupervised detection (Vos et al., 2022; Ergen & Kozat, 2020), leveraged autoencoder reconstruction errors for ECG anomaly detection (Roy et al., 2023), and utilized them in domains such as railway system monitoring (Wang et al., 2022). On the other hand, GCNs exploit the inherent graph structure of data for representation learning, such as integrating LOF for intrusion detection (Qin et al., 2025), combining with TCN for cloud server performance anomaly detection (Tan et al., 2024), and enabling weakly supervised video anomaly detection (Park et al., 2023). More recently, hybrid models that combine LSTMs and GCNs have shown effectiveness by jointly modeling temporal and spatial features, including applications in network traffic anomaly detection (Kaya et al., 2025), encrypted traffic classification (Yuan et al., 2025), and EEG signal analysis (Kang et al., 2026). Distinct from these works, our approach focuses on facial emotion embedding sequences extracted from video modality, treating them as anomalous signals for modeling and detection.

## 3 PROPOSED METHOD

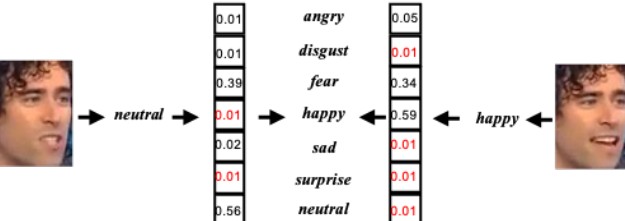

Figure 2: Emotional probability scores of facial categories.

Yang et al. (Yang et al., 2020) formulated deception detection as a binary classification task by analyzing emotional tag features (ETF) from facial frames and demonstrated its effectiveness on a single-scene dataset. Inspired by their work, we argue that directly using discrete emotion labels may overlook latent emotional cues. Recent deep learning–based facial emotion recognition methods (Georgescu et al., 2019; Serengil & Ozpinar, 2024) typically generate emotion probability matrices and determine the final category using a maximum-probability selection strategy, which may discard informative secondary emotions. As shown in fig.2, classifying a sample as "neutral" ignores a 0.2 probability of "happy", while labeling it as "happy" fails to reflect a 0.34 probability of "fear". Such hidden emotions and their temporal variations provide critical clues for deception detection. As illustrated in fig.3, our proposed FMGTranDD framework addresses this by integrating three core components: (i) automated face acquisition and feature encoding, (ii) a hybrid Time-LSTM-GCN module, and (iii) a composite contrastive loss. The following subsections describe each component in detail.

**Automatic Facial Capture and Facial Feature Coding.** Recent studies generally rely on complex feature engineering processing. For example, the method proposed by Zhu et al. (Zhu et al., 2025) needs to obtain the segmentation results of the head and limb regions in advance and use

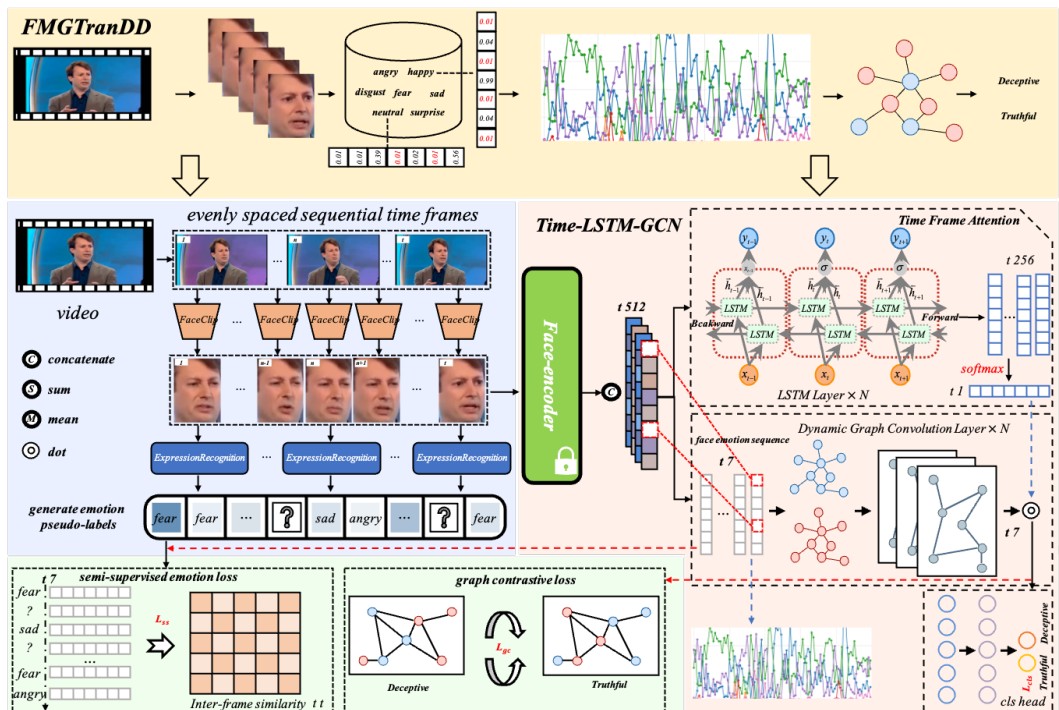

Figure 3: Automatic deception detection framework FMGTranDD model framework.

them as model input data after fine screening. This method has limitations in cross-scene migration ( such as not applicable to scenes where the limbs are not visible ), and will increase the cost of data acquisition and processing. For this reason, we have optimized and simplified the process. Since facial expression is the key carrier of emotional expression, it is necessary to accurately capture the facial area of the video subject. As shown in fig.3, given a video $V \in R^{T \times 3 \times H \times W}$ ; $t$ represents the number of video frames ; $H$, $W$ denotes the width and height of the video frame. According to the discrete uniform sampling, the index of t video frames is obtained, as shown in equation 1:

$$x_i = start + i \bullet \left[ \frac{end - start}{t - 1} \right], i \in \{0, 1, \cdots, t - 1\}, \tag{1}$$

where $start$ represents the starting frame index, $end$ represents the end frame index, and $t$ represents the number of frames collected. In order to avoid too many interference faces in the first and last clips of the video, we set the time interval as $start = T/10$, $end = T - T/10$ to effectively filter non-target faces. According to the calculated index value, the facial image data is collected in turn and the main emotional categories are labeled. The industrial-grade Deepface tool developed by Serengil et al. (Serengil & Ozpinar, 2024) integrates multiple types of face recognition and emotion recognition algorithms. We use its extract_faces and analyze functions to extract facial features and label the main emotion types in turn. During the data processing, when the face is not detected in the index frame, the preset compensation mechanism will be automatically triggered to ensure the integrity of the data acquisition. For example, when the face is not detected in the $x_i$ frame, the face of the $x_i - j$ or $x_i + j$ index is obtained by decreasing or increasing the strategy. When the analyze function or other emotion recognition methods cannot effectively recognize facial emotions, the sample is automatically marked as an ' unknown ' category. The FaceClip and ExpressionRecognition processes shown in fig.3 automatically complete facial cutting and emotion recognition. The emotion labels generated in this process are defined as ' emotion pseudo-labels '. The process is fully automated and does not require manual intervention frame selection.

Finally, each video generates a facial sequence $f \in R^{t \times 3 \times h \times w}$, where $h$ and $w$ represent the width and height of the facial image, which is uniformly set to $112 \times 112$ pixels. In order to extract effective facial features, we compared and evaluated the performance of various feature coding backbones, including traditional CNN convolution, CNN3D (Tran et al., 2018), ResNet (He et al., 2016), and

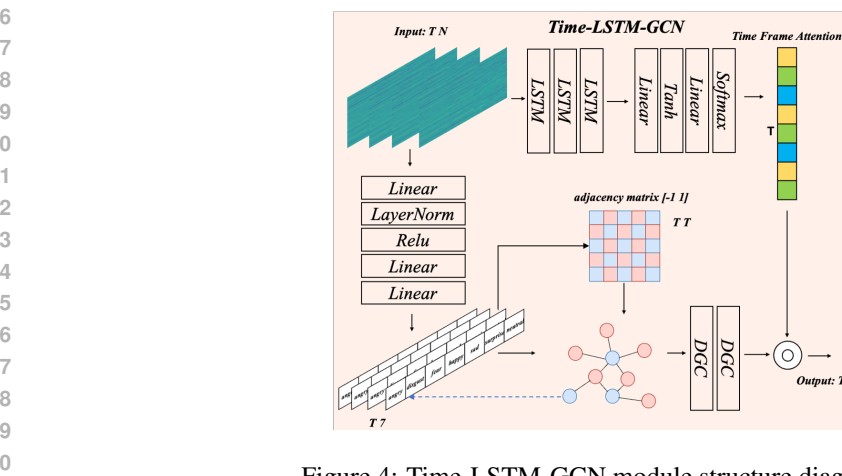

Figure 4: Time-LSTM-GCN module structure diagram.

TransFace (Dan et al., 2023). Finally, TransFace based on Transformer architecture is selected as the feature extractor. The pre-training model can directly extract facial features without fine-tuning. The output $f_t \in R^{t \times N}$ sequence processed by TransFace contains facial key detail feature information.

**Time-LSTM-GCN Hybrid Module.** The LSTM and GCN models have wide application value in various abnormal signal detection tasks. Inspired by the research of Kaya et al. (2025); Roy et al. (2023); Kang et al. (2026), we treat the extracted basic facial representation sequence $f_t$ as a special type of signal data for processing and analysis. In order to better mine the decision boundary between real and deceptive behavior in $f_t$, we designed the Time-LSTM-GCN module. As shown in fig.4, the Time-LSTM-GCN module is a two-branch structure. Inspired by the attention mechanism proposed in Hu et al. (2018); Qilong Wang & Hu (2020), the first branch of Time-LSTM-GCN uses the LSTM network to capture the inter-frame temporal correlation of the facial representation sequence $f_t$, and generates the spatio-temporal attention weight $T_a \in R^{t \times 1}$ after linear transformation. The weight matrix quantifies the importance of each facial frame in the deception detection task. As shown in equation 2-equation 3 :

$$f_i = LSTM\left(f_t\right)_i, \tag{2}$$

$$T_a = L\left(Tanh\left(L\left(Softmax\left(f_i\right)\right)\right)\right), \tag{3}$$

where $i$ represents the number of LSTM layers ; $L$ denotes the linear mapping layer ; $Tanh, Softmax$ is the corresponding activation function. The second branch first encodes the facial representation sequence $f_t$ into the facial emotion embedding sequence $f_e \in R^{t \times 7}$, as shown in equation 4:

$$f_e = L\left(LN\left(Relu\left(L\left(L\left(f_t\right)\right)\right)\right)\right), \tag{4}$$

where $LN$ represents layer normalization, and $Relu$ is the corresponding activation function. The facial emotion embedding sequence $f_e$ is regarded as an advanced signal, which reflects the changes of the main subjects ' facial display emotions and hidden emotions in the video. In order to mine the decision boundary between real and deceptive behaviors, we use dynamic graph convolution ( DGC ) to model the spatio-temporal relationship of $f_e$. In addition, this process will continuously update the node relationship of $f_e$. Specifically. The adjacency matrix is generated according to the neighbor relationship of $f_e$, as shown in equation 5:

$$\mathbf{M} = \mathrm{clamp}\left[\left(\frac{f_e}{\|f_e\|_2 + \epsilon}\right) \cdot \left(\frac{f_e}{\|f_e\|_2 + \epsilon}\right)^T, 1, -1\right], \tag{5}$$

where $clamp$ represents the numerical constraint function, and the numerical range is constrained between 1 and-1 ; $\|\cdot\|_2$ denotes L2 regularization, $\epsilon = 1e-8$ prevents division of 0. Further, $f_e$ and M are introduced into DGC to construct a spatio-temporal relationship. The emotional probability matrix of each frame is regarded as a node, and the node relationship is updated repeatedly through

multi-layer DGC. The specific representation process is shown in equation 6 :

$$G_j = Relu \left( f_e + \frac{L(f_e)}{clamp\_min \left[ \sum_i^7 f_e + \epsilon, 1.0 \right]} \bullet M \right)_j, \tag{6}$$

where $clamp\_min$ represents the minimum value of 1.0, and $j$ represents the number of DGC layers. Finally, we combine the information from both branches, $f_{end} = T_a G_j$. We pass $f_{end}$ into the classification head to get the probability scores of the real and deceptive pairs. The classification head consists of a simple linear mapping layer.

**Combined Opposition Loss.** In the previous section, we elaborated on the core module Time-LSTM-GCN of the framework. The module takes the basic facial representation sequence $f_t$ as the signal input, transforms it into a discriminative facial emotion embedding sequence by learning mapping, and establishes its spatio-temporal correlation features. In order to enhance the effect of spatio-temporal relationship modeling, we specially design a combined opposition loss function to improve the performance of the method by simultaneously optimizing the decision boundaries of real and deceptive behaviors.

Semi-supervised emotional loss. The actual meaning expressed by the facial emotion embedding sequence is the seven emotion scores of each frame. Therefore, we designed a semi-supervised loss to improve the representation ability of facial emotion embedding sequences. Specifically, based on the available pseudo-labels generated in the previous section, we calculate the cross-entropy loss for data with valid sentiment annotations. As shown in equation 7 :

$$\mathcal{L}_{sup}\left(\dot{p_{ij}}, \dot{y_{ij}}\right) = -\frac{1}{T} \sum_{i=1}^{T} \sum_{j=1}^{C} \left[ \dot{y_{ij}} \bullet log\left(\dot{p_{ij}}\right) \right], \tag{7}$$

where $\dot{p_{ij}}$ is the multi-class emotion prediction probability of the $i$ th frame of the model ; $\dot{y_{ij}}$ is the effective real emotion label ; $T$ is the number of frames ; $C$ is the total number of emotional categories, defaulted to 7. Secondly, inspired by the research of Dwibedi et al. (Qilong Wang & Hu, 2020), considering the high correlation of adjacent expression features, we use the tridiagonal mask constraint model to learn the smooth transition of adjacent frame features, so that it conforms to the continuity prior of expression changes. Specifically, as shown in equation 8-equation 9 :

$$\mathcal{L}_{\text{unsup}}(p_{ij}) = -\frac{1}{T} \sum_{i=1}^{T} \log \left( \frac{\sum_{k=1}^{T} \exp(s_{tk}/\tau) \cdot M_{tk}}{\sum_{k=1}^{T} \exp(s_{tk}/\tau)} \right), \tag{8}$$

$$M_{tk} = \begin{cases} 1, & \text{if } |t-k| \le 1, \\ 0, & \text{otherwise.} \end{cases}, \tag{9}$$

where $p_{ij}$ is the multi-class emotion prediction probability of the i th frame of the model ; $s_{tk} = \cos(h_t, h_k)/\tau$, $h_t$ is the L2 normalized feature of the $t$ frame, and $cos$ refers to the cosine similarity function; $\tau$ is the temperature scaling coefficient, which enlarges the similarity difference ; $M_{tk}$ is a tridiagonal matrix, the main diagonal and adjacent 1 positions are 1, and the rest are 0. The total emotional semi-supervised loss is shown in equation 10:

$$\mathcal{L}_{ss}\left(p_{ij}, y_{ij}\right) = sup\mathcal{L}_{unsup}\left(p_{ij}, y_{ij}\right) + unsup\mathcal{L}_{unsup}\left(p_{ij}\right), \tag{10}$$

where sup and unsup are the loss control coefficients, defaulted to 1.0 and 0.4.

Graph contrastive loss. In order to further optimize the decision boundary, we draw on the ideas of various graph comparison learning methods (Xu et al., 2024; Liu et al., 2024; Zhang et al., 2023). The facial emotion embedding sequence $f_e$ output by the DGC module is constructed as a graph structure data, in which each frame emotion probability matrix is used as a graph node. By comparing the similarity between the nodes in the same sequence ( positive samples ) and the nodes between different sequences ( negative samples ), the model can simultaneously learn the local structure and global difference characteristics of the emotional embedding sequence. In the specific implementation, firstly, the anchor nodes are randomly selected. The positive samples come from

the random nodes of the same sequence, and the negative samples come from the random nodes of other sequences. The specific comparison loss is shown in equation 11:

$$\mathcal{L}_{\text{gc}} = -\frac{1}{N} \sum_{i=1}^{N} \log \left( \frac{\exp(s_i^+/\tau)}{\exp(s_i^+/\tau) + \exp(s_i^-/\tau)} \right), \tag{11}$$

where $s_i^+$, $s_i^-$ are positive and negative sample similarity respectively ; $\tau$ is the temperature coefficient ; $N$ is the effective contrast logarithm.

Total loss. Combined with semi-supervised emotional loss and graph comparison loss, the total loss is as shown in equation 12 - equation 13 :

$$\mathcal{L}_{cls}(p_i, y_i) = -\sum_{i=1}^{C} [y_i log(p_i)], \tag{12}$$

$$\mathcal{L}_{total} = \mathcal{L}_{cls} + \alpha \mathcal{L}_{ss} + \beta \mathcal{L}_{gc}, \tag{13}$$

where $mathcalL_{cls}$ is a common cross-entropy loss used to calculate the label difference between true and deceptive ; $p_i$ is the probability of predicting truth or deception ; $y_i$ is a true or deceptive label, $\alpha, \beta$ is the loss control coefficient default to $1/T$ and 1.

## 4 EXPERIMENTALS

**Datasets.** Commonly used datasets for deception detection include DOLOS (Guo et al., 2023), Bag-of-Lies (Gupta et al., 2019), RLT (Pérez-Rosas et al., 2015), and SEUMLD (Xu et al., 2025). DOLOS originates from a game show and contains 1,675 video clips (899 deceptive, 776 truthful), while Bag-of-Lies is collected in laboratory settings with 325 clips (162 deceptive, 163 truthful). Both are relatively large and suitable for model evaluation. In contrast, RLT (121 clips) and SE-UMLD (76 participants with multimodal data) are limited in scale. Therefore, our experiments are primarily conducted on DOLOS and Bag-of-Lies.

**Implementation Details.** We implemented our method using the PyTorch framework and trained it on an NVIDIA A6000 GPU. The model was optimized with the AdamW optimizer for approximately 300 epochs with a learning rate of 1e-4. Following the evaluation protocol in Zhu et al. (2025), we report accuracy (ACC), F1 score (F1), and area under the ROC curve (AUC) as performance metrics.

Table 1: Comparison of deception detection performance of different methods on the DOLOS and BagofLies datasets (evaluation metrics are ACC, F1, and AUC)

| Method | DOLOS (Guo et al., 2023) | | | Bag-of-Lies (Gupta et al., 2019) | | |
|---|---|---|---|---|---|---|
| | ACC | F1 | AUC | ACC | F1 | AUC |
| TimeSformer (Bertasius et al., 2021) | 0.6649 | 0.6642 | 0.7352 | 0.5296 | 0.4455 | 0.5407 |
| VideoMAEv2 (Wang et al., 2023) | 0.6740 | 0.7174 | 0.7080 | 0.5231 | 0.3130 | 0.5211 |
| VideoMamba (Li et al., 2024) | 0.6914 | 0.7358 | 0.7287 | 0.5816 | 0.4256 | 0.4569 |
| LieNet (Karnati et al., 2022) | 0.5650 | 0.6972 | 0.5102 | 0.5978 | 0.5814 | 0.5809 |
| FacialCueNet (Nam et al., 2023) | 0.6098 | 0.6865 | 0.6199 | 0.5623 | 0.6326 | 0.5953 |
| PECL (Guo et al., 2023) | 0.6475 | 0.7120 | 0.6271 | 0.5951 | 0.5106 | 0.5941 |
| GLDD (Kang et al., 2024) | 0.5547 | 0.6252 | 0.5213 | 0.5666 | 0.5517 | 0.5789 |
| DLF-BRAM (Zhu et al., 2025) | **0.7886** | **0.8227** | 0.7772 | 0.6050 | 0.5563 | 0.6107 |
| **Proposed** | 0.7870 | 0.7866 | **0.8285** | **0.6667** | **0.6667** | **0.6380** |

**Contrast Experiment.** In this subsection, we compare our proposed FMGTranDD with DLF-BRAM (Zhu et al., 2025), LieNet (Karnati et al., 2022), FacealCueNet (Nam et al., 2023), PECL (Guo et al., 2023), GLDD (Kang et al., 2024), as well as representative video understanding approaches, including TimeSformer (Bertasius et al., 2021), VideoMAEv2 (Wang et al., 2023), and VideoMamba (Li et al., 2024). Among them, LieNet integrates facial and audio features; PECL combines facial, textual, and body-motion cues; DLF-BRAM fuses facial and body-motion features; while FacealCueNet and GLDD rely solely on facial features. Following the protocol in (Zhu

Table 2: Ablation results (ACC, F1, AUC) on the DOLOS dataset using different backbones, spatiotemporal modeling modules, and loss function configurations.

| | Module | | | Loss | | | ACC | F1 | AUC |
|---|---|---|---|---|---|---|---|---|---|
| Backbone | Only-LSTM | Only-GCN | Time-GCN-LSTM | $\mathcal{L}_{cls}$ | $\mathcal{L}_{ss}$ | $\mathcal{L}_{gc}$ | | | |
| CNN3D (Tran et al., 2018) | | | | ✓ | | | 0.7229 | 0.7220 | 0.7738 |
| CNN2D | ✓ | | | ✓ | | | 0.7470 | 0.7465 | 0.7420 |
| ResNet-152 (He et al., 2016) | ✓ | | | ✓ | | | 0.7952 | 0.7953 | 0.8517 |
| | ✓ | | | ✓ | | | 0.8133 | 0.8128 | 0.8171 |
| | | ✓ | | ✓ | | | 0.8313 | 0.8313 | 0.8763 |
| TransFace (Dan et al., 2023) | | ✓ | | ✓ | ✓ | | 0.8554 | 0.8553 | 0.8803 |
| | | ✓ | | ✓ | ✓ | ✓ | 0.8614 | 0.8615 | 0.8943 |
| | | | ✓ | ✓ | ✓ | ✓ | **0.8675** | **0.8675** | **0.9158** |

et al., 2025), we adopt the corresponding train/validation/test splits for evaluation. On the DOLOS dataset, three predefined train–test splits are used, and we report the average performance across three metrics. On the Bag-of-Lies dataset, we adopt a 7:1:2 split for training, validation, and testing. The results are summarized in Table 1.

On the DOLOS dataset, FMGTranDD achieves the best AUC, significantly outperforming all competing methods. This demonstrates the model's superior ability to learn clearer decision boundaries and discriminative features. The improvement primarily stems from the Time-LSTM-GCN module, which effectively models the spatiotemporal dependencies of facial emotion embeddings, together with the contrastive loss, which enforces local and global discriminative constraints. Although DLF-BRAM slightly outperforms FMGTranDD in ACC and F1, it relies on multi-modal inputs (facial + body motion), whereas our method achieves comparable or superior performance using only a single modality, highlighting its efficiency and robustness. In contrast, general-purpose video models perform noticeably worse on DOLOS, indicating that while such models excel in generic video understanding, they fail to capture the subtle facial variations critical for deception detection. In the Bag-of-Lies dataset, FMGTranDD consistently achieves the best results across all three metrics, outperforming all baselines. Notably, multi-modal methods such as LieNet, PECL, and DLF-BRAM perform worse than single-modality approaches, suggesting that when domain shifts and modality-specific noise are present, simple feature fusion does not guarantee performance gains and may even degrade results. By contrast, FMGTranDD, through spatiotemporal modeling of facial expression embeddings, effectively captures micro-expression conflicts and suppression patterns commonly associated with deceptive behavior, leading to stronger generalization and cross-dataset robustness. Meanwhile, general-purpose video models achieve no more than 0.58 ACC on Bag-of-Lies, further confirming that such models favor global semantic representations but fail to encode the abnormal cues required for deception detection. In summary, the comparative experiments validate the advantages of FMGTranDD in terms of discriminative power, generalization, and single-modality efficiency. They also underscore the importance of task-specific modeling of abnormal signals for deception detection, rather than relying solely on generic video understanding frameworks.

**Ablation Experiment.** To assess the contribution of different components in our proposed framework, we conducted systematic ablation studies on the DOLOS dataset. These experiments covered the impact of the backbone, spatiotemporal modeling modules, and loss functions, as well as two key hyperparameters: the number of input frames and the number of DGC layers. The results are summarized in Tables 2–4. As shown in Table 2, the choice of backbone and model structure plays a critical role in overall performance. When trained with only the classification loss, both CNN3D and CNN2D exhibit limited discriminative ability. Replacing them with ResNet-152 significantly boosts the AUC to 0.8517, while leveraging the face-recognition–pretrained TransFace (112×112) further strengthens representation quality, confirming that high-quality facial features are essential for deception detection. Building on this foundation, introducing spatiotemporal modeling leads to substantial improvements: using LSTM alone raises the AUC to 0.8763, and further combining it with GCN improves ACC/F1 to 0.8554/0.8553 and AUC to 0.8803. With the full Time-GCN-LSTM architecture, all metrics reach new highs, indicating that temporal dependencies and graph-structured relationships are complementary for deception detection. In addition, fig.5 visualizes the raw features and those processed by the Time-LSTM-GCN module. In the unmodeled raw features (a) and (c), genuine and deceptive samples exhibit significant overlap with blurred inter-class boundaries. After applying Time-LSTM-GCN (b) and (d), the features form clearer clus-

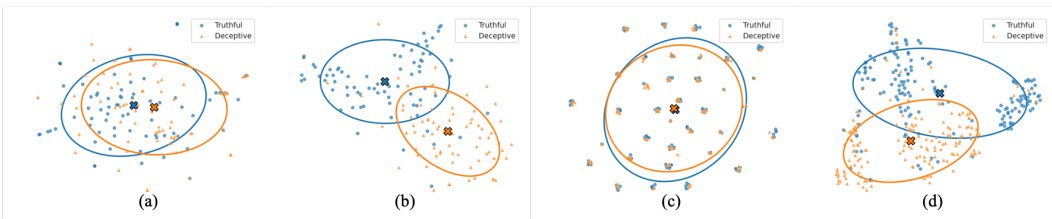

Figure 5: Feature distribution visualization before and after applying the Time-LSTM-GCN module on different datasets. (a) Raw feature distribution of the DOLOS dataset; (b) DOLOS features encoded by Time-LSTM-GCN; (c) Raw feature distribution of the Bag-of-Lies dataset; (d) Bag-of-Lies features encoded by Time-LSTM-GCN.

Table 3: Effect of different input frame lengths on DOLOS dataset (ACC, F1, AUC).

| Frames | ACC | F1 | AUC |
|---|---|---|---|
| 8 | 0.8434 | 0.8434 | 0.8420 |
| 16 | 0.8675 | 0.8674 | 0.9109 |
| 24 | 0.8434 | 0.8434 | 0.9092 |
| 32 | **0.8675** | **0.8675** | **0.9158** |

Table 4: Effect of DGC layer depths on DOLOS dataset (ACC, F1, AUC).

| DGC Layers | ACC | F1 | AUC |
|---|---|---|---|
| 2 | **0.8675** | **0.8675** | **0.9158** |
| 4 | 0.8313 | 0.8308 | 0.8803 |
| 6 | 0.7952 | 0.7953 | 0.8636 |
| 8 | 0.7892 | 0.7893 | 0.8529 |

ters in the emotional embedding space, and the separation between truthful and deceptive samples is substantially enhanced. This also shows that Time-LSTM-GCN can effectively capture the spatiotemporal dependencies of sequences and increase the distance between different categories in the latent space, thereby optimizing the decision boundary. The ablation on loss functions further highlights their necessity. With only the classification loss, the model shows limited discriminative power. Adding the semi-supervised loss $\mathcal{L}_{ss}$ increases the AUC to 0.8803, demonstrating its effectiveness in enhancing emotion embedding representations. Incorporating the graph-contrastive loss $\mathcal{L}_{gc}$ provides additional gains, pushing AUC to 0.9158 and ACC to 0.8675, the overall best. These results suggest that semi-supervised and contrastive learning are complementary, jointly improving both local representation quality and global decision boundaries, thereby enhancing robustness and generalization.

As shown in Tables 3–4, both the number of input frames and the depth of DGC layers substantially influence performance. Increasing the number of frames generally improves results, though not monotonically. With only 8 frames, the AUC is limited to 0.8420, indicating that short sequences fail to capture complete facial dynamics and miss deception-related cues. Expanding to 16 frames yields a marked improvement, suggesting this length adequately covers a full cycle of facial emotion changes. However, at 24 frames, performance drops, although the AUC remains high (0.9092). Finally, with 32 frames, performance rises again to the overall best, showing that longer windows can still provide marginal gains in decision boundaries, albeit with diminishing returns. For DGC layers, the best performance is observed with 2 layers. As the depth increases, performance steadily declines, reflecting the effects of over-smoothing and overfitting, which reduce the discriminative capacity of embedding sequences. Thus, a moderate depth (e.g., 2 layers) strikes the best balance, effectively modeling graph relations while preserving feature distinctiveness.

## 5    CONCLUSION

This paper proposes FMGTranDD, which models the spatio-temporal relationships of facial emotion embeddings and incorporates adversarial optimization strategies to enhance the recognition of anomalous signals. The method not only demonstrates strong discriminative and generalization abilities in deception detection but also provides a feasible pathway for broader anomaly behavior recognition tasks. In the future, this approach can be extended to emotion recognition and behavioral analysis of patients with depression and other mental disorders, enabling the capture of subtle facial dynamics and emotional abnormalities, and offering new technical support for mental health monitoring and clinical-assisted diagnosis.

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

## A APPENDIX

### A.1 ETHICS STATEMENT

We have ensured that all data used in this research comply with ethical guidelines and proper citations. The study follows responsible AI practices and avoids any data manipulation or misrepresentation.

## A.2 REPRODICIBILITY STATEMENT

The experiments in this work are fully reproducible. All datasets and code used for model training and evaluation are publicly available. Detailed information on the experimental setup, model configurations, and hyperparameters is provided to enable independent reproduction of results.

## A.3 USE OF LLMS

This work does not involve the use of any large language models (LLMs), such as GPT, BERT, or similar models.

