# OpenReview forum: "FMGTranDD: A Deception Detection Method Based on Spatiotemporal Facial Abnormal Emotional Changes"
_ICLR.cc/2026/Conference — ICLR 2026 Conference Withdrawn Submission_

### Official Review · Reviewer_qQGj · 2025-10-24

**Soundness:** 3
**Presentation:** 3
**Contribution:** 2
**Rating:** 4
**Confidence:** 4

**Summary:**

This paper proposes FMGTranDD, a unimodal (video-only) deception detection method. The method first encodes facial frames using a pre-trained Transformer (TransFace), and then models the spatio-temporal relationships within this sequence using the proposed Time-LSTM-GCN module. The decision boundary is optimized via a combined loss function comprising a semi-supervised emotion loss and a graph contrastive loss. Experiments on the DOLOS and Bag-of-Lies datasets show that the method outperforms several multimodal and unimodal baselines.

**Strengths:**

A key strength of this work lies in its effective integration of multiple components to tackle deception detection as an abnormal signal recognition task based on facial emotional changes. The framework successfully combines a Transformer-based feature extractor (TransFace), a novel Time-LSTM-GCN module specifically designed to model spatiotemporal relationships within facial emotion embedding sequences, and a well-designed combined loss function incorporating semi-supervised emotion learning and graph contrastive objectives. This synergistic combination enables the model to effectively capture relevant spatiotemporal patterns indicative of deception, leading to strong empirical results that outperform several baseline methods on benchmark datasets.

**Weaknesses:**

1.The paper claims to follow the split protocol from TITS on Bag-of-lies dataset. However, it actually is a 7:1:2 ratio split, whereas the TITS protocol is a person-based split. Besides, all the compared methods likely adhere to this person-based split. Using a different split strategy could lead to an incorrect comparison.

2.While the overall framework is new, the individual components are largely adaptations of existing techniques. The use of Transformer encoders, LSTM, GCN and loss is standard. The novelty is limited.

**Questions:**

See weakness

---

### Official Review · Reviewer_ZdHZ · 2025-10-29

**Soundness:** 2
**Presentation:** 2
**Contribution:** 1
**Rating:** 2
**Confidence:** 5

**Summary:**

This paper addresses deceptive behavior detection by framing it as an abnormal signal recognition problem, aiming to capture deviations from normal behavioral patterns. The approach first converts facial data from videos into learnable facial emotion embedding sequences. A Time-LSTM-GCN module is then proposed to model the spatiotemporal relationships among these embeddings. The model is trained with a combined adversarial loss, which (1) uses semi-supervised learning to enhance emotion representation and (2) encourages the model to distinguish local intra-sequence structure and global inter-sequence differences. Experiments demonstrate that the proposed baseline outperforms several methods.

**Strengths:**

- The method maps video frames to facial emotion embeddings (7 basic emotions: 6 of Ekman and also neutral I assume). Prior deception detection work rarely uses a structured, sequence-based emotion embedding representation in this way.
- The Time-LSTM-GCN module and graph contrastive loss are aimed to be primarily applied to the emotion embeddings, not raw visual or multimodal features. In this way, the system is learning patterns of emotional dynamics over time, which is more interpretable than purely raw features.

**Weaknesses:**

1)	The paper frames deception detection as an anomaly detection problem, implicitly assuming that deceptive behavior is “rare.” However, at the individual level, this assumption may not hold. Some individuals may lie frequently or exhibit behaviors that deviate from the population norm, making the rarity assumption unreliable. Therefore, treating deception as an anomaly without accounting for personal behavioral baselines could limit the method’s validity and generalizability.

2)	The comparison tables (e.g., Table 1) in the paper may be misleading because the listed methods do not always use the same modalities or cues. For example, some methods incorporate audio features, while others rely solely on visual signals. A more fine-grained comparison, distinguishing methods by the types of cues used, would provide a more accurate and meaningful evaluation.

3)	L62-67, no citations, but claims require citations.

4)	Low-cost? How did you measure this (I do not see any analysis in this regard in the paper)? Why is it needed in deception detection?

5)	The technical novelty appears limited. The main components: Transformer-based feature extraction, LSTM-GCN for spatiotemporal modeling, and adversarial loss optimization, are well-established techniques in affective computing and behavioral analysis. The work mainly combines existing methods in an application-oriented manner rather than introducing fundamentally new algorithmic ideas. A clearer justification of what constitutes the genuine methodological innovation would strengthen the paper. I further have some more in-depth questions about each component as follows.

6)	The authors’ justification for focusing exclusively on the video modality is not fully convincing. While they argue that multimodal approaches suffer from interpretability issues and potential cross-modal interference, deception is a multifaceted psychological and behavioral phenomenon that typically manifests across multiple modalities (e.g., facial expressions, voice tone, speech content, physiological cues). Relying solely on visual information risks overlooking critical deceptive indicators and oversimplifying the problem. A more rigorous justification for excluding other modalities, or at least a discussion of the trade-offs, would strengthen the study’s rationale.

7)	The proposed facial capture and feature coding pipeline offers a practical and automated preprocessing approach, but its methodological novelty is limited. It heavily relies on off-the-shelf models (DeepFace, TransFace) and simplistic uniform sampling, with little evidence of how these choices impact deception-related signal quality. The use of emotion pseudo-labels introduces potential noise and bias, while the assumption that deceptive cues align with categorical emotion labels is questionable. Stronger justification, validation, or ablation studies are needed to demonstrate that this pipeline captures meaningful deception-related features rather than general facial expressions.

8)	The adjacency matrix $M$ is built from normalized dot products of feature vectors ($f_e$), clamped to $[-1, 1]$. This forms a basic similarity graph rather than a learned or dynamically optimized one. No details are provided on how the dynamic update occurs or whether it is trained end-to-end. Moreover, the use of clamp functions and arbitrary thresholds introduces non-differentiable or ad-hoc components, which may limit learning stability. Further details are needed in such a manner.

9)	The connection between emotion embeddings and deception-specific relationships is speculative. The method assumes that deceptive behavior manifests as distinct spatio-temporal emotion patterns. Without explicit evidence that emotion graphs correlate with deceptive cues, the model risks functioning as a generic temporal graph framework rather than one tailored to deception analysis.

10)	The final fusion $f_{\text{end}} = T_a \cdot G_j$ is an element-wise product, with no theoretical or experimental justification for this choice, nor comparison to alternatives such as concatenation or attention-based fusion. This simplistic design may limit expressiveness and cause loss of complementary information between branches.

11)	The graph convolution update formula (Eq. 6) mixes normalization, nonlinear activation, and adjacency multiplication in a non-standard manner, making the method difficult to reproduce and lacking formal justification. Can you explain it in detail?

12)	The experimental analysis is also very limited in terms of datasets. Many studies in this field use many more datasets, including cross-dataset analysis, too. Instead, this paper primarily relies on experimental analysis of the DOLOS dataset, including ablations performed solely on that dataset. It is not clear if the findings are equally valid for the Bag-of-Lies dataset.

13)	The combined opposition loss is conceptually interesting, integrating semi-supervised emotion supervision and graph contrastive learning to model spatio-temporal relationships. However, it still relies heavily on pseudo-labels, which may introduce noise, and assumes that emotion embeddings reliably correlate with deception, an assumption not empirically validated. While ablation studies demonstrate the contribution of each loss component, the random sampling strategy for contrastive loss remains ad hoc, and the practical advantage over simpler approaches may depend on dataset characteristics.

14) Some ablations are missing:
a) Effect of attention weights ($T_a$).
b) Fusion strategy (element-wise product vs alternatives).
c) Quality of pseudo-labels vs. ground truth emotion labels (if supplied).

**Questions:**

Listed in the Weaknesses part.

---

### Official Review · Reviewer_r2G4 · 2025-10-30

**Soundness:** 3
**Presentation:** 3
**Contribution:** 2
**Rating:** 4
**Confidence:** 3

**Summary:**

This paper proposes a deception detection method called FMGTranDD, which is based on spatiotemporal abnormal facial emotion changes. The paper treats deceptive behavior as an abnormal signal recognition problem, aiming to capture abnormal features from regular behavioral patterns. Its core contributions are as follows:
1.Convert the facial information in videos into learnable facial emotion embedding sequences.
2.Propose a Time-LSTM-GCN module to model the spatiotemporal relationships of these embedding sequences.
3.Design an optimization strategy that incorporates adversarial loss, including semi-supervised emotion loss and graph contrastive loss, to enhance the decision boundary.
Experiments show that this method outperforms existing multimodal or unimodal methods on the DOLOS and Bag-of-Lies datasets. The paper emphasizes the efficiency of unimodal methods and provides reproducible code.

**Strengths:**

The paper demonstrates strengths in several aspects:
1.The Time-LSTM-GCN module, which integrates spatiotemporal attention, represents a novel design for deception detection. Additionally, the integration of semi-supervised learning and contrastive learning into adversarial loss enhances the model’s representational capability.
2.The experiments are comprehensive, with performance validated across multiple datasets. A systematic ablation study is also conducted (as shown in Tables 2-4).
3.The framework is described intuitively, and Figure 3 visualizes the overall workflow, facilitating understanding of the method.

**Weaknesses:**

1.The experiments are only based on Western datasets (e.g., DOLOS) and do not test cross-cultural scenarios. The method relies on facial visibility, so it may not be applicable to scenarios with body occlusion.
2.The comparative experiments are insufficient: the comparison with general video models (e.g., VideoMAE) is superficial, and there is no in-depth discussion on why task-specific models are more superior.
3.In preprocessing, unrecognizable emotions are simply labeled as the "unknown" category, which may introduce noise. The paper does not elaborate on how these "unknown" samples are handled during training and inference, yet they might contain important deceptive cues that the model has not yet learned to identify.

**Questions:**

The experimental validation of this paper is entirely based on datasets from Western cultural contexts (e.g., DOLOS). Given the significant cultural differences in facial emotion expression, what is the generalization ability of this method when applied to non-Western cultural groups (e.g., East Asia, the Middle East)? Can the authors provide evidence of its effectiveness in cross-cultural scenarios?
This method is highly dependent on clear, unobstructed facial images. How does the model's performance degrade in more realistic and complex scenarios—such as partial occlusion, large-angle pose variations, or low lighting? The paper does not conduct any evaluation or discussion on this aspect.

---

### Official Review · Reviewer_siz2 · 2025-10-30

**Soundness:** 3
**Presentation:** 4
**Contribution:** 2
**Rating:** 4
**Confidence:** 3

**Summary:**

This paper proposes FMGTranDD, a framework for video-based deception detection that models spatiotemporal facial emotional changes as an abnormal signal recognition task. The method generates emotion embedding sequences from video frames and processes them with a "Time-LSTM-GCN" module, which combines an LSTM for temporal modeling and a Dynamic Graph Convolution (DGC) for spatial relationship modeling. The model is trained with a composite loss function that includes a classification loss, a semi-supervised emotion loss, and a graph contrastive loss. The paper reports state-of-the-art performance on the DOLOS and Bag-of-Lies datasets and provides ablation studies to support its design choices.

**Strengths:**

1. Clear Problem Formulation and Motivation: The paper is well-structured and presents a clear motivation for treating deception detection as an abnormal signal recognition problem based on irregularities in emotional expression. This conceptual framing is sound and provides a coherent rationale for the work.

2. Use of Rich Emotion Embeddings: The decision to use the full probability distribution of facial emotions as embeddings, rather than discrete labels, is a positive choice. This allows the model to capture more nuanced and subtle emotional cues, such as the presence of conflicting emotions, which is a relevant and valuable aspect for the task.

**Weaknesses:**

The core technical components of the model are not novel. The combination of an LSTM and a GCN to process sequential graph-structured data is a well-established pattern in the literature. The paper fails to clearly articulate what is new about the "Time-LSTM-GCN" module beyond this standard combination. The use of terms like "adversarial loss" for a standard contrastive loss and "dynamic" for a cosine-similarity-based graph further overstates the technical contribution and gives the impression of repackaging existing techniques.

**Questions:**

1. The "Time-LSTM-GCN" module combines two well-known components (LSTM and GCN). Can the authors explicitly state what the novel architectural or algorithmic contribution of this module is, beyond the straightforward application of existing techniques? How does this work advance the state of the art in model design, as opposed to its application to a new problem?

2. The graph is constructed from emotion probability vectors, which are the output of a classification model. This means errors in the emotion recognition step will directly corrupt the graph structure. Have the authors analyzed the robustness of the model to noise or errors in the initial emotion labeling? Could a model that operates directly on raw facial features learn a more robust and effective representation than one that depends on this intermediate, potentially unreliable, symbolic representation?

3. The ablation study in Table 4 shows a significant performance drop when using more than 2 GCN layers, which is a classic sign of over-smoothing. This suggests the GCN component is not effectively learning to propagate useful information. Could the authors provide evidence that the GCN is learning meaningful graph-level patterns, rather than acting as a shallow non-linear transformation? For instance, are the learned adjacency matrices interpretable, or do they show any structure beyond random noise?

---

### Note · Authors · 2025-11-19

I have read and agree with the venue's withdrawal policy on behalf of myself and my co-authors.